# Native structure of mosquito salivary protein uncovers domains relevant to pathogen transmission

Shiheng Liu [1,2,4], Xian Xia [1,2,4], Eric Calvo [3] & Z. Hong Zhou [1,2]✉

Female mosquitoes inject saliva into vertebrate hosts during blood feeding. This process transmits mosquito-borne human pathogens that collectively cause ~1,000,000 deaths/year. Among the most abundant and conserved proteins secreted by female salivary glands is a high-molecular weight protein called salivary gland surface protein 1 (SGS1) that facilitates pathogen transmission, but its mechanism remains elusive. Here, we determine the native structure of SGS1 by the *cryoID* approach, showing that the 3364 amino-acid protein has a Tc toxin-like Rhs/YD shell, four receptor domains, and a set of C-terminal daisy-chained helices. These helices are partially shielded inside the Rhs/YD shell and poised to transform into predicted transmembrane helices. This transformation, and the numerous receptor domains on the surface of SGS1, are likely key in facilitating sporozoite/arbovirus invasion into the salivary glands and manipulating the host's immune response.

Mosquito-borne diseases have shaped the history of humankind by altering the outcomes of major wars from ancient Athens to World War II[1]. Some mosquito species, including *Aedes aegypti*, *Culex pipiens*, and *Anopheles gambiae*, are key vectors of mosquito-borne pathogens, including arthropod-borne (arbo) viruses such as dengue, yellow fever, West Nile, Zika and chikungunya, as well as the malaria-causing parasite *Plasmodium falciparum*. Approximately one million people die of vector-borne diseases every year at present (https://www.who.int/news-room/fact-sheets/detail/vector-borne-diseases) and an astonishing estimate of half of human deaths since the dawn of humanity can be linked to mosquitoes[1]. Pathogen transmission occurs during blood feeding when an infected female mosquito injects saliva into a vertebrate host[2]. Component analyses of mosquito saliva via transcriptional, proteomic and functional studies have shown that salivary molecules have anti-hemostatic and immuno-modulatory properties relevant for blood feeding acquisition. Mosquito saliva facilitates blood feeding by inhibiting platelet aggregation, blood coagulation, and vasoconstriction[3]. Saliva and salivary gland proteins have also been shown to enhance the severity of transmitted diseases[4–7]. Arbovirus transmission through infected mosquito bites or needle

injections with uninfected mosquito bites led to more severe disease than virus transmission solely through needle injection[8,9]. Mosquito saliva also enhances *Plasmodium falciparum* infectivity and malaria disease progression[10].

Among the estimated 100–200 proteins in mosquito saliva, 30–40% belong to previously uncharacterized protein families and have unknown functions[11]. One of the most abundant salivary proteins in *Aedes aegypti* mosquitoes is a high molecular weight (>300 kDa) protein called salivary gland surface protein 1 (SGS1)[12]; it is thought to have been acquired via horizontal transfer from bacterial endosymbionts[13–15]. SGS1 is exclusively expressed in the salivary glands of adult female mosquitoes, which suggests that SGS1 is involved in blood-feeding and pathogen transmission[15]. Screening of monoclonal antibodies enriched for recognition of salivary gland surface epitopes revealed that SGS1 is required for invasion of *Aedes aegypti* salivary glands by *Plasmodium gallinaceum* sporozoites[15,16]. Reverse genetic approaches by RNA interference (RNAi) and CRISPR/Cas9 further confirmed the role of SGS1 in facilitating sporozoite invasion[17]. Zika virus transmission was also positively affected by SGS1, likely by a similar mechanism[18]. SGS1 orthologs, including a ~200 kDa protein

[1]Department of Microbiology, Immunology, and Molecular Genetics, University of California, Los Angeles, CA 90095, USA. [2]California NanoSystems Institute, University of California, Los Angeles, CA 90095, USA. [3]Laboratory of Malaria and Vector Research, National Institute of Allergy and Infectious Diseases, National Institutes of Health, Rockville, MD 20852, USA. [4]These authors contributed equally: Shiheng Liu, Xian Xia. ✉e-mail: Hong.Zhou@UCLA.edu

with neutrophil chemotactic activity from *Anopheles stephensi* saliva[19] and a ~387-kDa protein with immunomodulatory properties from *Aedes aegypti* saliva[20], are thought to enhance pathogenicity of arboviruses and *Plasmodium* parasites by modulating the host's immune response[12].

Efforts to determine the SGS1 structure are hindered by difficulties in generating properly modified and folded SGS proteins through a recombinant approach. Bioinformatic analyses have also been inconclusive because most sequences do not possess readily identifiable domains. The only known domain information currently available is that SGS1 contains rearrangement hotspot (Rhs) or tyrosine-aspartate (YD)-repeats preceding a panel of multi-pass transmembrane helices[15]. However, SGS1 has been found in both saliva and the basal lamina of the medial and distal lateral salivary gland lobes – both soluble and non-membranous environments[15]. Bioinformatic analyses show that SGS1 lacks a classical signal peptide; furthermore, the differing masses of SGS1 orthologs in saliva and acinar cells suggest that SGS proteins may undergo proteolytic cleavage prior to secretion into the salivary duct[12].

Here, we utilize the *cryoID* approach[21] to directly image and identify native proteins in *Aedes aegypti* salivary gland extract and have determined the atomic structure of native SGS1 by cryogenic electron microscopy (cryo-EM). The structure revealed that the predicted transmembrane (TM) helices are partially folded and fully embedded in a Tc toxin-like Rhs/YD shell, explaining how SGS proteins exist in soluble environments. A combination of structural comparison with phylogenetic and sequence analyses uncovered a previously unidentified cleavage site of an aspartic protease, which reconciles the large body of existing biochemical data and suggests a mechanism for transforming and releasing the putative TM helices. These helices and numerous receptor domains resolved in the structure likely facilitate sporozoite/arbovirus invasion into the salivary gland or manipulate the host's immune response.

## Results

### Cocoon-shaped architecture of SGS1

Cryo-EM images were obtained from salivary gland extracts without fractionation. SDS-PAGE and proteomic analyses revealed that SGS1 (Uniprot accession: AAEL009993-PA) and SGS1b (Uniprot accession: AAEL009992-PA) were among the most abundant proteins present in the salivary gland extract (~24% for SGS1; ~12% for SGS1b), and the only ones that had molecular weights (MW) larger than 188 kDa (Supplementary Fig. 1a, b). Accordingly, images of both negatively stained and frozen-hydrated samples showed particles of similar size and shape (Supplementary Fig. 1c). 2D averages from the cryo-EM images confirmed that the predominant species have a dimension of ~200 Å by ~100 Å (Supplementary Fig. 1c). By combining these 2D classes for in-depth 3D analysis, we obtained a cryo-EM map at an overall resolution of 3.3 Å, enabling us to identify this particle as SGS1 through *cryoID*[21] (Supplementary Figs. 1d and 2) and to model amino acid (aa) 1 to 3042 of the full-length, uncleaved SGS1, which is comprised of 3364 residues (Fig. 1a, Supplementary Fig. 3 and Supplementary Movie 1). We were also able to build two N-glycan sites N59 and N1149 into this native structure of SGS1. The structure suggests that SGS1 in the salivary gland is likely cleaved at the C terminus, consistent with our mass spectrometry (MS) results (Supplementary Fig. 4a) and a previous observation[17] showing that no peptides after residue 3035 of SGS1 were recovered from salivary glands in *Aedes aegypti*.

The SGS1 structure is cocoon shaped, with dimensions of 210 Å × 115 Å × 86 Å (Fig. 1b, c). It is organized into 6 domains: two β-propeller domains, a rearrangement hotspot/tyrosine-aspartate (Rhs/YD)-repeats domain, a carbohydrate-binding module (CBM), a lectin carbohydrate-recognition domain (lectin-CRD), and a wedge domain (Fig. 1a, b and Supplementary Movie 1). The C-terminal moiety, a ~230 aa-long sequence previously predicted to form a set of TM helices[15], is

surprisingly almost fully buried within the chamber inside the cocoon shell and forms daisy-chained helices, which extend across the entire internal space to an opening at the middle of the Rhs/YD-repeats domain (red in Fig. 1c).

Sequence analysis showed shared domain organization among homologous SGS proteins in mosquitoes (Supplementary Fig. 5). Phylogenetic analysis revealed that SGS1 homologs are present across *Aedes*, *Culex* and *Anopheles* genera (Fig. 1d), all recognized as encompassing species that spread severe human diseases. By contrast, homologous SGS proteins in most bacteria are much shorter, only containing part of the Rhs/YD-repeats domain. Our observation that the architecture of SGS1 is conserved within pathogen-vector mosquitoes but divergent across other species is consistent with the role of SGS1 in blood feeding or disease transmission[15–18].

### Rhs/YD-repeats and β-propeller 2 form a large SGS1 chamber

The Rhs/YD-repeats domain is built by large β-sheets consisting of an enormous number (~100) of β-strands folded into a left-handed spiral of almost four turns, creating a hollow shell i.e., Rhs/YD shell) (Supplementary Movie 2). The bottom of this Rhs/YD shell is extended by the β-propeller 2 domain, forming an internal chamber ~30 Å wide and ~110 Å long (two left panels of Fig. 2a). The top of the Rhs/YD shell is plugged by the C-terminal internal spiral (green in Fig. 2a), a conserved sequence known as the Rhs repeat-associated core (Rhs core, InterPro accession: IPR022385). In the middle of the Rhs/RD shell, a large opening between spiral layers 1 and 2 was observed. The short sequence from residues 1303 to 1321 folds into a 3-turn helix with an extended loop that bisects the opening (labeled as "fence" in the right panel of Fig. 2a).

DALI search of the Rhs/YD-repeats domain identified two top-ranking folds: Tc toxin and teneurin. Side-by-side comparison of these structures revealed that the SGS1 Rhs/YD-repeats domain is nearly identical to that in the BC components of bacterial Tc toxins[22,23] and highly similar to that of eukaryotic teneurins[24,25] (Fig. 2b), though the latter has a much shorter N-terminal β-sheet (cyan in Fig. 2b). The SGS1 chamber encapsulates the C-terminal moiety of ~230 residues. This is close to the embedded ~200 C-terminal residues of the Tc toxins but different from that of teneurins, which only accommodates a C-terminal moiety of ~90 residues. The increased cargo capacity of SGS1 and Tc toxins is consistent with their larger shell dimensions as compared to the shell of teneurins (Fig. 2b).

The Rhs core represents a conserved sequence shared among SGS1, Tc toxins, and teneurins[22,26]. In Tc toxin, it functions as an aspartyl auto-protease (AP) to free the toxic C-terminal domain from the rest of the protein[22], while that in teneurin does not[24]. Sequence alignment indicated that SGS proteins and Tc toxin contain all three residues required for catalysis (one Arg followed by two catalytic Asp), but teneurin lacks the last catalytic Asp, suggesting that SGS proteins are likely the AP with the corresponding auto-cleavage site located between residues 2733-2734 in SGS1 (Fig. 2c). Indeed, mass spectrometry shows that an SGS1 peptide covering this auto-cleavage site was present in the salivary gland extract but was not detected in the saliva of *Aedes aegypti* (Supplementary Fig. 4), indicating an SGS1 AP cleavage event: the cleavage has already happened in saliva but not yet in the salivary gland. Comparison of the Rhs core structures of SGS1 with Tc-toxin and teneurin indicated that SGS1 has a higher level of similarity to Tc-toxin than to teneurin in terms of overall fold. Nonetheless, the last catalytic residue of SGS1 (Asp2729) is pulled farther away from the catalytic center than that (Asp674) in Tc toxin (Fig. 2d). These observations suggest that a major conformational change must occur in SGS1 to activate its catalytic activity during its secretion from the gland to saliva.

### The putative TM helices are only partially folded in the chamber

Right after the putative AP cleavage site, a ~230 aa long C-terminal moiety (residues 2734-2966) was previously thought to form a set of

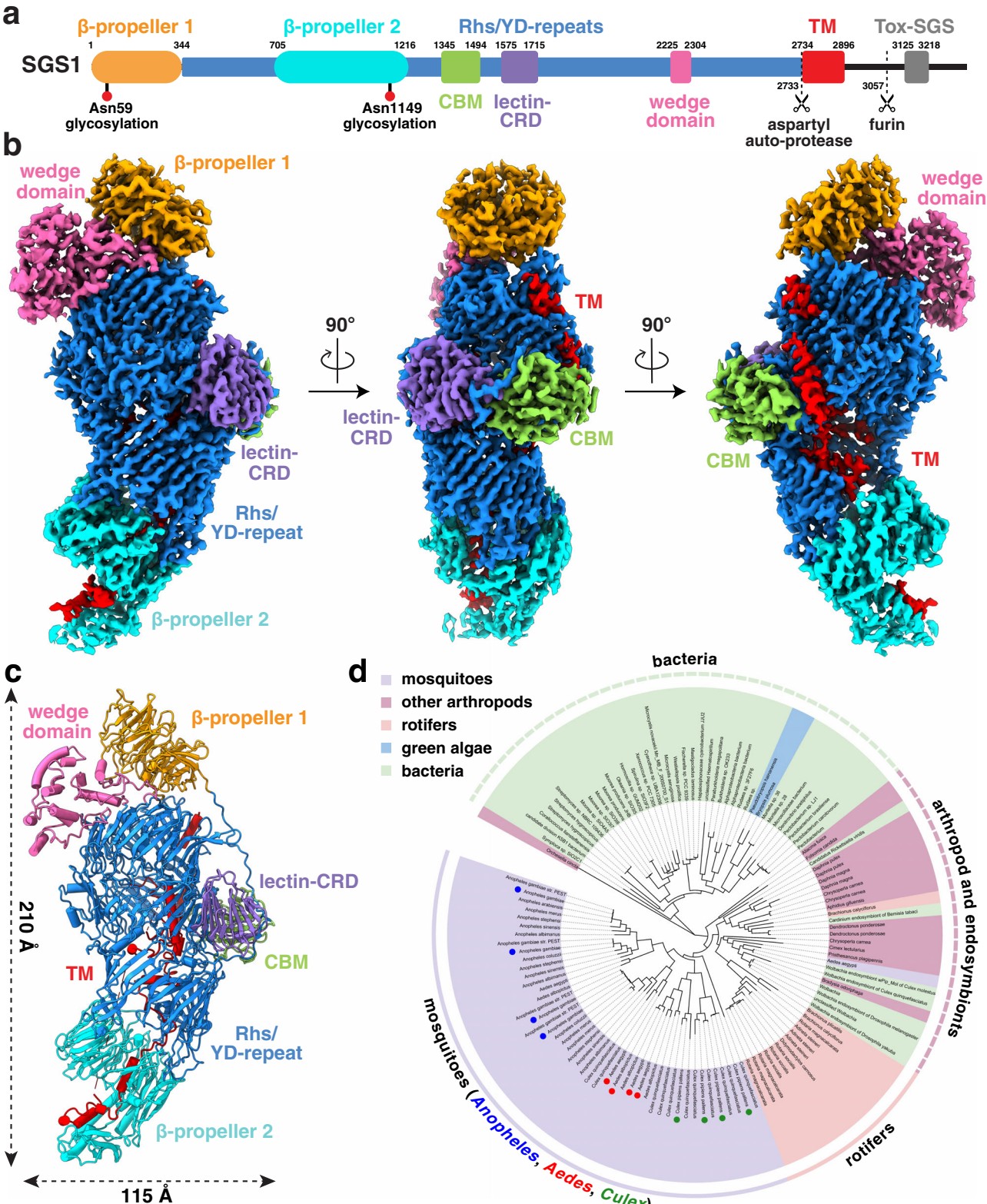

**Fig. 1 | Overall architecture of the salivary gland surface protein 1 (SGS1) from *Aedes aegypti*. a** Domain diagram of SGS1. Residue numbers at domain boundaries are indicated. Two putative protease cleavage sites are shown as dashed line with scissors. Two N-glycan sites are shown as red circles. Abbreviated domain names: CBM (carbohydrate-binding module), lectin-CRD (lectin carbohydrate-recognition domain), TM (putative transmembrane helices), Tox-SGS (salivary gland secreted protein domain toxin). **b** Different views of the cryo-EM density map of SGS1. **c** Atomic model of SGS1 shown in cartoon representation. **d** Maximum-likelihood phylogenetic analysis of SGS1 using IQ-TREE. Targets selected for further multiple sequence alignment are labeled as red (*Aedes*), blue (*Anopheles*) and green circles (*Culex*).

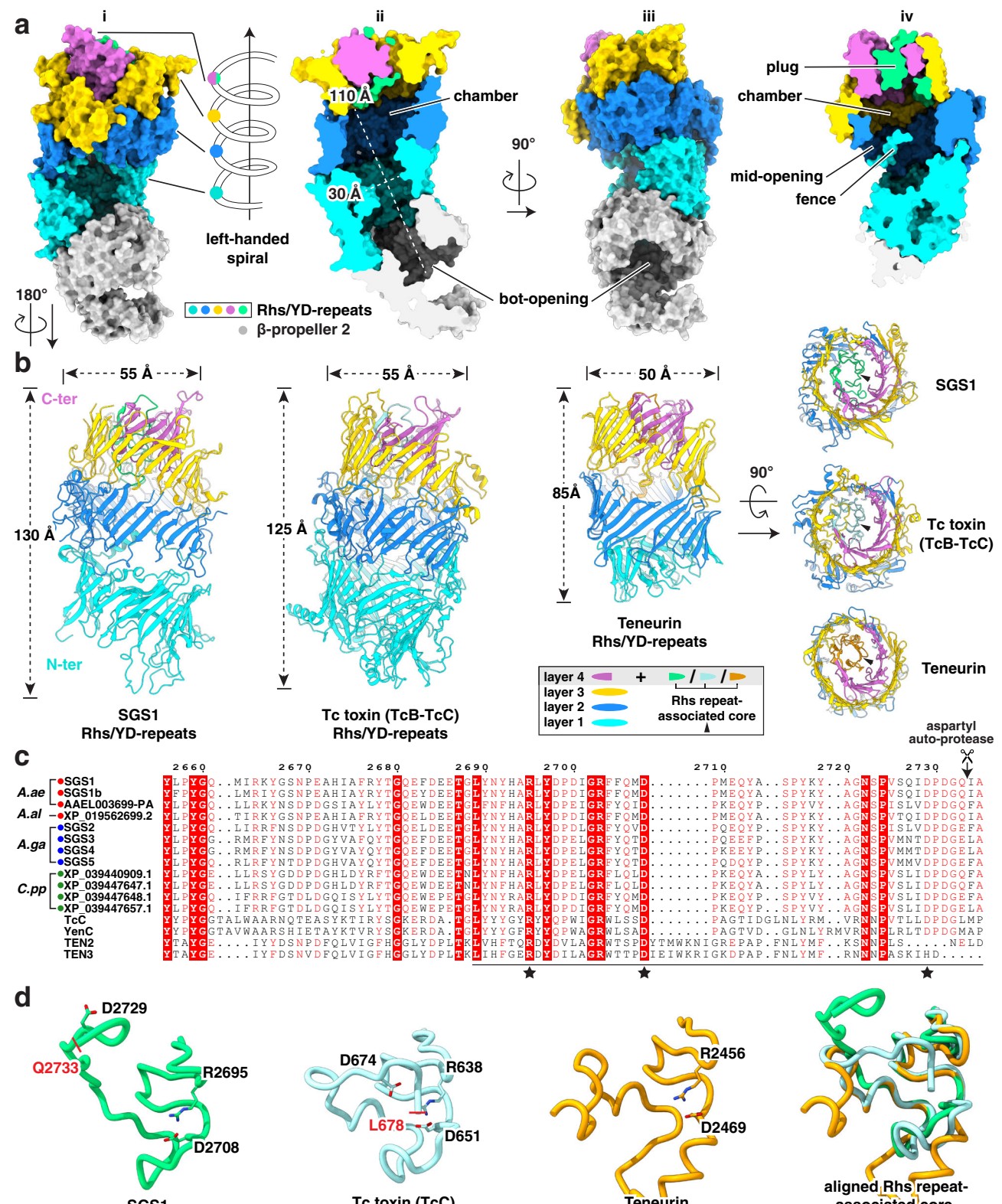

**Fig. 2 | Structure of the Rhs/YD-repeats domain. a** Molecular surface of SGS1 Rhs/YD shell extended by the β-propeller 2 domain (panel i or iii) and its slices (panel ii or iv). From N to C-terminus, the left-handed, approximately four turns are colored as cyan, blue, yellow and purple-green, respectively. The Rhs repeat-associated core (Rhs core) of SGS1 is colored as green. **b** Structural comparison of the Rhs/YD shells between SGS1 and its homologs. Color scheme is the same as (**a**), except the Rhs cores of Tc-toxin and teneurin are colored as turquoise and orange, respectively. **c** Multiple sequence alignment of the Rhs core. The first twelve sequences are

mosquito SGS proteins derived from the selected targets in Fig. 1d; TcC and YenC are the C component of bacteria Tc toxins; TEN2 and TEN3 are eukaryotic teneurins. The putative cleavage site of aspartyl auto-protease (AP) is indicated with an arrow and scissors. The positions of the three residues required for catalysis are labeled with stars. **d** Structural comparison of partial Rhs cores [underlined in (**c**)] among SGS1, Tc toxin C component and teneurin. The AP cleavage sites in SGS1 (putative) and TcC are shown as short red lines.

TM helices[15], which bear moderate sequence similarity (E value: 2.37e-05) to aa permeases. Surprisingly, however, this segment is almost fully embedded inside the SGS1 chamber as follows: it travels for over 120 Å, from the central region of the Rhs/YD shell between spiral layers 2 and 3, all the way to the β-propeller 2 domain; it then travels almost all the way back to exit the shell wall from the mid-opening of the SGS1 structure (Figs. 3a and 2a, and Supplementary Movie 2). This C-terminal moiety, consisting mostly of α-helices and loops, has extensive interaction with the interior of the Rhs/YD-repeats and the β-propeller 2 domain, largely via hydrophobic contacts (Fig. 3b). A notable exception is that residues 2865-2868 of the moiety form a short β-strand and associate with the inner β-sheet of spiral layer 1 via main-chain interactions.

To reconcile the cryo-EM structure with previous sequence predictions, we performed in-depth sequence analysis for this C-terminal moiety. Our prediction indeed showed six TM helices with high confidence score (Fig. 3c). However, among these predicted TM helices, numbers 1, 3, 4, 5 and 6 did not fully match the secondary structures resolved in the cryo-EM structure, while helix number 2 was not resolved at all (Fig. 3a, c), indicating that these predicted TM helices are only partially folded and entirely sheltered in the chamber in the cryo-EM SGS1 structure. Notably, the sequence of these predicted TM helices was highly conserved among the SGS proteins of different mosquito species, suggesting an essential function of the predicted TM helices that is shared by all SGS proteins.

Immediately afterwards, the remaining sequence of SGS1 (residues 2967-3042) traverses through the shell wall and then folds into two additional helices, one of which runs along the crevice between spiral layers 2 and 3 and the other along the crevice between layers 3 and 4 (Figs. 3a and 2a). Notably, this external segment of these two helices and the internal segment of the predicted TM helices are connected by a loop through the mid-opening of the Rhs/YD shell. This type of connection through a mid-opening is shared with teneurins but not with Tc-toxin. In Tc-toxin, the encapsulated C-terminal toxin exits from the shell through the bottom gate (labeled as bot-gate in Fig. 3d). In contrast to their conserved Rhs-YD shells, the C-terminal moiety of SGS1 (residues 2734-3042) exhibited no sequence similarity with that of Tc-toxins and teneurins, in line with their diverse functions.

Residues after 3042 primarily contained sequence predicted to form a conserved, arthropod-specific domain, commonly referred to as *salivary gland secreted protein domain toxin* (Tox-SGS)[27] (Supplementary Fig. 5). Upstream of the Tox-SGS, sequence alignment of SGS homologs identified one single furin cleavage site R-X-K-R conserved in SGS proteins from *Aedes* and *Culex* mosquitoes and a corresponding weak furin cleavage site located nearby in each SGS homolog from *Anopheles* mosquitoes (Fig. 3e, f and Supplementary Fig. 5). In SGS1, the furin cleavage site is located right after residue 3057. No densities for Tox-SGS were observed in the cryo-EM map, which indicates that the furin cleavage has already occurred in SGS1 in salivary gland extract. This result is consistent with our mass spectrometry result (Supplementary Fig. 4a) and previous report[17] showing that no peptides after residue 3035 were recovered from salivary glands in *Aedes aegypti*.

### Receptor domains on the Rhs/YD shell

The exterior surface of the SGS1 Rhs/YD shell is decorated by five domains: β-propeller 1 (residues 1–344), β-propeller 2 (residues 705–1216), a carbohydrate-binding module (CBM: residues 1345–1494), a lectin carbohydrate-recognition domain (lectin-CRD: residues 1575-1715), and a wedge domain (residues 2225–2304, 2326–2465) (Fig. 4a). None of these domains have been annotated based on sequence analysis, and the wedge domain is a previously unidentified fold, based on search results from the DALI server. Except for the β-propeller 1 domain, the other four domains are formed by sequence segments inserted into the sequences of the Rhs/YD-repeats domain (Fig. 1a).

The β-propeller 1 domain resembles the canonical β-propeller structure. It has seven symmetrically arranged blades with a closed bottom and sits atop the C-terminal side of the Rhs/YD-repeats domain (Fig. 4a, b). The top side of β-propeller 1 interacts with spiral layer 4 of the Rhs/YD shell via blades 5 and 6 (Fig. 4b) and associates with residues 2225–2304 of the wedge domain via loops in blades 1, 2, 3 and 7 (Fig. 4c). The wedge domain (residues 2225–2304, 2326–2465) wedges between the Rhs/YD shell and the β-propeller 1 domain, tilting the axis of the latter ~45° away from the longitudinal axis of the shell wall (Fig. 4c). The wedge domain contains two conserved disulfide bonds (Supplementary Fig. 5) that can be broken in a reducing environment to trigger possible dislocation of the β-propeller 1 domain. Naturally prevalent, seven-bladed propeller proteins can mediate transient protein-protein interactions via their top, bottom, and side face[28] and can also bind various ligands, including carbohydrates, via their side face[29,30].

Unlike the β-propeller 1 domain, the β-propeller 2 domain is located on the other end of the Rhs/YD shell and differs from the canonical β-propeller in three major aspects: distorted arrangement of its nine blades, insertion of a protruding lid atop the blades, and an occupied central tunnel (Fig. 4d). Traversing through this central tunnel is the middle portion of the daisy-chained helices formed by the sequence predicted to fold into TM helices (residues 2809–2853, a few residues before 2809 that were not modeled due to weak densities) (Fig. 3c, e). Within the tunnel, the α-helix containing residues 2839-2853 lines a hydrophobic cleft formed by blades 6, 7 and 8, squeezing out blade 7, and makes the β-propeller appear asymmetric (Fig. 4e). The lid is formed by two fragments: the first (residues 899-942) is a helix-loop-helix fragment inserted between blades 5 and 6; the other (residues 993–1068) comprises a three-stranded β-sheet and an α-helix, inserted between blades 6 and 7 (Fig. 4d). The lid engages in hydrophobic interactions with the rest (residues 2809–2838) of the middle portion of the daisy-chained helices.

In the middle of the Rhs/YD shell are the two putative polysaccharide binding domains, CBM and lectin-CRD, both resembling an immunoglobulin (Ig)-like β-sandwich, which consists of two sheets with antiparallel β-strands (Fig. 4a, f, g). Despite low sequence identity (9%) between CBM and lectin-CRD, their structures have a root-mean-square deviation of 3.0 Å across 98 Cα atom pairs (Fig. 4h). CBM and lectin-CRD are positioned close to each other (Fig. 4a), potentially promoting cooperative binding, as demonstrated by 10- to 100-fold enhancement in ligand binding affinity with two clustered carbohydrate-binding sites[31,32].

In total, SGS1 possesses four putative receptor domains − two β-propellers, one CBM and one lectin-CRD − that could mediate protein-protein interactions and/or facilitate carbohydrate binding. Neural network-based modeling with four representative SGS proteins of various mosquito species indicated that such receptor domains are common among them (Supplementary Fig. 6). Similarly, the TcB component of Tc toxin has a β-propeller for TcA binding[23,33], while the Rhs/YD shell of teneurin possesses the NHL, Lphn1-lectin and Lphn1-olf domains as receptor domains to mediate interactions with other proteins (Fig. 4i)[34].

## Discussion

Pathogen transmission through mosquitoes involves sophisticated biological processes (*i.e.*, pathogen acquisition to midgut via blood feeding, systemic infection of mosquito tissues, salivary gland penetration, and saliva injection) compared with direct contact transmission or indirect airborne or vehicle-borne transmission. As such, our understanding of the underlying mechanisms remains limited as compared to other modes of transmission. In this study, we have determined the native structure of SGS1 from a mosquito

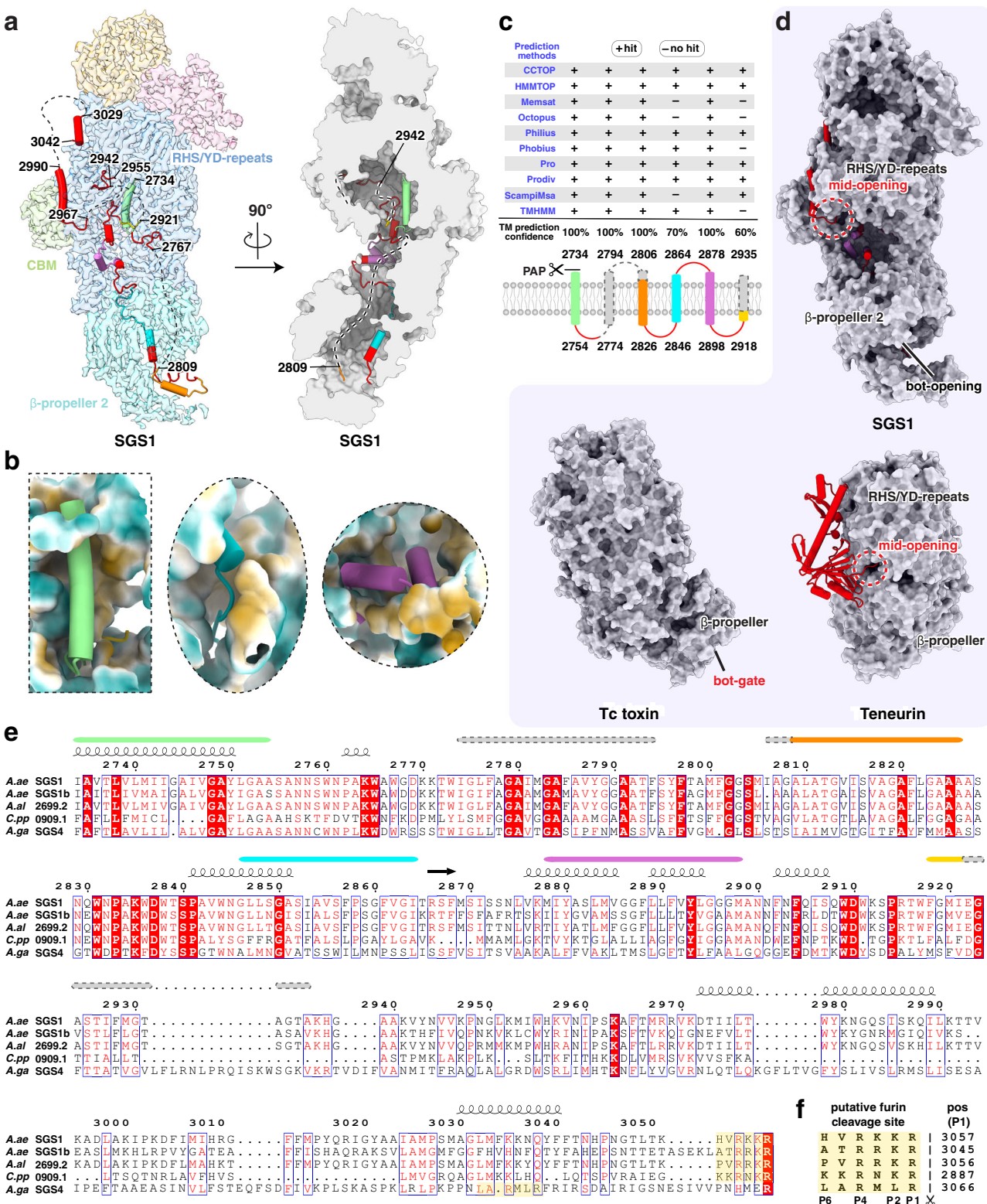

**Fig. 3 | Putative transmembrane helices of SGS1. a** Transparent map density of SGS1 (left panel) and the surface slice (right panel) showing daisy-chained helices predicted to form transmembrane helices. The flexible, unmodeled regions are indicated with dashed lines. **b** Close-up view of the daisy-chained helices engaged in hydrophobic patches. **c** Prediction of transmembrane helices in SGS1 using CCTOP server[56]. The confident score was hit rate considering all the prediction methods. The unmodeled regions are indicated as dashed lines and dashed rectangles. **d** Comparison of the surface opening/gate in SGS1, Tc toxin and teneurin. These three structures are aligned based on their Rhs/YD shell and displayed separately. **e** Multiple sequence alignment of the C-terminal moieties in mosquito SGS proteins. Two layers of secondary structure are shown. The upper layer represents the prediction result shown in (**c**); the lower layer is derived from our cryo-EM structure. The putative furin cleavage sites are colored with transparent yellow. **f** The putative furin cleavage sites are aligned and indicated using residue numbers at their P1 position (the cleavage occurs right after P1).

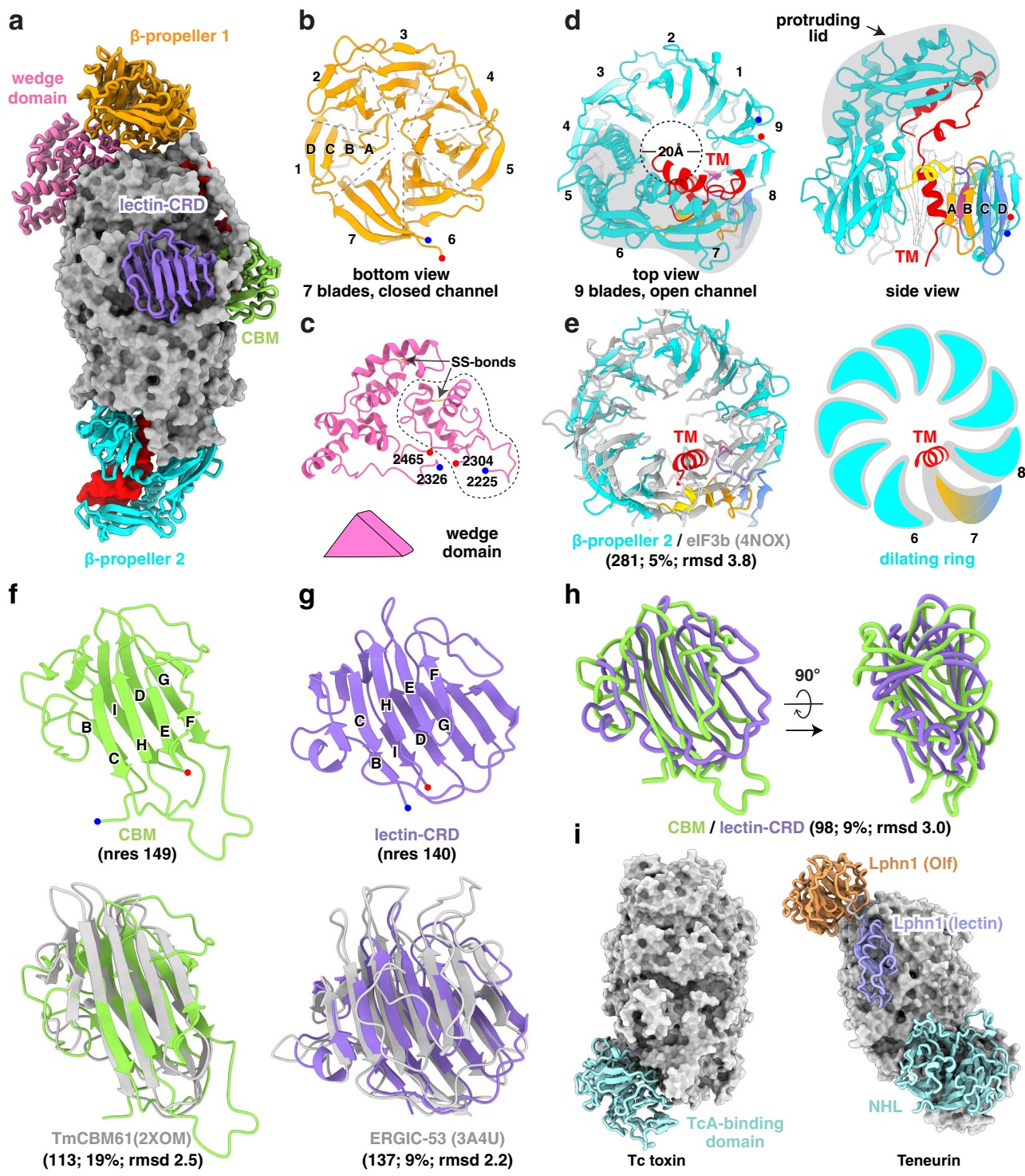

**Fig. 4 | Receptor-binding domains of SGS1. a** The receptor-binding domains (shown as cartoon) decorating the Rhs/YD shell (shown as gray surface). **b** Symmetric seven-bladed β-propeller 1 viewed from the bottom side. Its N-terminus and C-terminus were labeled as blue and red dots, respectively. The four β-strands from N- to C-terminus in one blade are referred to as A, B, C, D. **c** Structure of the wedge domain containing residues 2225-2304 (dashed circle) and 2326-2465. Two conserved disulfide bonds (SS-bonds) are shown in yellow. **d** Different views of the asymmetric nine-bladed β-propeller 2 interacting with

predicted transmembrane fragments (red). **e** Left panel: structural comparison of the β-propeller 2 with symmetric nine-bladed eIF3b (PDB ID: 4NOX). They share a sequence identity of 5% with a RMSD of 3.8 Å across 281 Cα atom pairs. Right panel: a schematic representation of the comparison. **f, g** Upper panel: the immunoglobulin (Ig)-like β-sandwiches of CBM (**f**) and lectin-CRD (**g**). Lower panel: structural comparison of these Ig-fold domains (green and purple) with their homologs participating in polysaccharide recognition (grey). **h** Different views of the superimposed CBM/lectin-CRD. **i** The receptor domains in Tc toxin and teneurin.

salivary gland. We show that the 3364-aa protein has a Tc toxin-like Rhs/YD shell, four receptor domains and a set of putative embedded TM helices within the Rhs/YD shell during the soluble conformation of this large protein accompanying mosquito-borne pathogen

transmission. The central questions emerging from our study concern the potential transformation of the daisy-chained helices inside the Rhs/YD shell and the possible functions of the numerous receptor domains.

Considering that Tc toxins possess the same AP cleavage site for delivering the C-terminal toxin, we speculate that the AP site in SGS1 is responsible for freeing the daisy-chained helices (*i.e.*, the predicted TM helices) from the Rhs/YD shell. Our mass spectrometry results revealed that, in the soluble environment of mosquito saliva, auto-cleavage of SGS1 has already occurred (Supplementary Fig. 4); thus, the putative TM helices must remain associated with the shell wall at this stage. The hydrophobic patches on the inner surface of the shell wall provide the microenvironment to harbor these helices and shield them from their external soluble surroundings (Fig. 3b). Such shielding resembles the hidden cytotoxic C component of the Tc toxin complex and may explain why these segments can be expressed without adversely impacting the cell. Notably, the daisy-chained helices inside the SGS1 shell wall are not yet folded into the predicted TM helices; these intermediate, metastable and extended structures might facilitate their detachment from the inner shell and egress through the opening as in Tc toxin[35]. These helices still reside within the Rhs/YD shell of SGS1 in saliva and could, after exposure to the host environment, detach during injection of SGS1-containing saliva in a mosquito blood meal. Possible detachment triggers include pH changes, host cofactors-binding and/or membrane attachment. Structural rearrangement near the middle opening of Rhs/YD shell (such as movement of the "fence" shown in panel iv of Fig. 2a) may be needed to free these daisy-chained helices.

Different from the highly symmetrical, disc-like structure of a canonical β-propeller, the nine-bladed β-propeller 2 domain in SGS1 has a large protruding lid and three distorted blades. The lid and distorted blades interact with the daisy-chained helices mainly through hydrophobic patches. Architectural rearrangement of the β-propeller has been observed in Tc toxin, where a distorted, five-bladed β-propeller refolds into a symmetrical, six-bladed β-propeller to translocate its toxic enzyme[33]. Structural comparison between SGS1 β-propeller 2 with a nine-bladed symmetrical β-propeller suggests that blades 6, 7 and 8 of β-propeller 2 may approach the central tunnel after releasing the putative TM helices, accompanied by relocation of the protruding lid.

In addition to the embedded TM helices, we identified four putative receptor domains (two β-propellers, one CBM and one lectin-CRD) on the shell surface of SGS1 (Fig. 4). Such receptor domains are common among SDS proteins of disease-transmitting mosquito species (Supplementary Fig. 6 and Fig. 4) and are potential mediators for protein-protein interactions and/or carbohydrate binding occurring during pathogen transmission; thus, they may serve as modules of SGS proteins that facilitate *Plasmodium* sporozoite/arbovirus invasion of the salivary gland and/or modulating the host's immune response.

Our results support the following model for the cleavage and secretion of salivary SGS proteins during blood feeding. Initially, the C-terminal Tox-SGS is cleaved by furin protease before being secreted to the basal lamina of the medial and distal lateral lobes, the major site for concentrating SGS proteins in the salivary gland[12,15]. Consistently, Tox-SGS was not identified in our cryo-EM structure or MS results. However, we are unable to exclude the possibility that other proteases may function to cleave Tox-SGS, since the furin sites of SGS proteins in *Anopheles gambiae* were not as remarkable as those of the SGS proteins in *Aedes* and *Culex* mosquitoes (Supplementary Fig. 5). The *Plasmodium* sporozoites/arbovirus may invade the salivary gland with help from the remaining SGS1 fragment in the medial and distal lateral lobes, likely via the receptor domains on the surface of the SGS1 protein. Next, the remaining SGS1 fragment is secreted into saliva when SGS1 cleavage catalyzed by its aspartyl auto-protease has already occurred (Supplementary Fig. 4). Previous western blot analyses revealed that a 300 kDa fragment of SGS proteins in the salivary gland of *Anopheles gambiae* is processed into a slightly less massive form prior to expulsion with the saliva[12]. The cryo-EM structure suggests that the C-terminal segment containing the putative TM helices remains fully embedded inside the SDS1 Rhs/YD shell, which protects the hydrophobic residues from the

soluble environment. During blood feeding, the remaining SGS1 fragment is injected into the host environment along with saliva, where SGS1 may interact with host cell via its receptor domains and release its putative TM helices from its Rhs/YD shell, potentially modulating host immune responses to benefit pathogen transmission.

## Methods

### Mosquito salivary gland dissection and saliva collection

*Aedes aegypti* (Liverpool strain) mosquitoes were reared at either the Department of Entomology and Fralin Life Science Institute, Virginia Tech or the Laboratory of Malaria and Vector Research, NIAID, NIH, under standard conditions (27 °C, 80% humidity, with a 12 h light/dark cycle). Sugar-fed female adult mosquitoes (5–7 days old) were anesthetized with $CO_2$, transferred to an ice-chilled plate, and their salivary glands (50 pairs) were dissected under a stereomicroscope in PBS (137 mM NaCl, 2.7 mM KCl, 4.3 mM Na2HPO4, and 1.4 mM KH2PO4, pH 7.4). Salivary gland extract (SGE) was obtained by disrupting the gland walls by sonication (Branson Sonifier450, Danbury, CT, USA) and cleared by centrifugation (12,000 g for 5 min at 4 °C). The supernatants were stored at -80 °C until used. Oil-induced saliva was collected as previously described with few modifications[36]. Briefly, alive mosquitoes were put on sticky tape with their back. Mosquito mouthparts were placed into 10 µl pipette tips with mineral oil and salivation was either passive or forced (injection of 200 nL of 3.6 mg/ml pilocarpine intrathoracically). After salivation, pipette tips with oil and saliva droplets were transferred to a tube containing10 µl of PBS, and the aqueous phase was obtained using centrifugation.

### Mass spectrometry

Mass spectrometry was performed as previously described[37,38]. Salivary gland extracts and saliva samples were subjected to mass spectrometry at Research and Technology Branch (NIAID, NIH). Samples from female *Aedes aegypti* were reduced in buffer containing 50 mM HEPES, pH 8.0, 10% acetonitrile and 5 mM DTT at 37 °C for 40 min. After cooling to room temperature, the samples were supplemented with iodoacetamide at a final concentration of 15 mM. After 15 min of alkylation, 200 ng of trypsin were added, and the samples were incubated at 37 °C for 15 h in a final volume of 40 µL. The solution was evaporated to near dryness under vacuum at 50 °C. 25 µL of 0.1% trifluoroacetic acid was added, and the pH was adjusted to 2.5 by adding 10% trifluoroacetic acid. Samples with an estimated protein content of less than 2 µg were desalted and concentrated with C18 µZip tips. Samples containing up to 10 µg of protein were desalted with C18 OMIX 10 solid phase extraction tips. The digests were eluted with 0.1% TFA, 50% acetonitrile and dried under vacuum. The peptides were dissolved in 12 µL 0.1% formic acid, 3% acetonitrile, and submitted to the LC-MS analysis using Orbitrap Fusion mass spectrometer (Thermo Fisher Scientific, West Palm Beach, FL, US) connected with an EASY nLC 1000 liquid chromatography system. Nano-LC was carried out with a 5 µL injection onto a PepMap 100 C18 3-µm trap column (2 cm, ID 75 µm; Thermo Fisher Scientific) and a 2 µm PepMap RSLC C18 column (25 cm, ID 75 µm; Thermo Fisher Scientific). The LC was operated at a 300 µL/min flow rate with a 100 min linear gradient from 100% solvent A (0.1% formic acid, and 99.9% water) to 40% solvent B (0.1% formic acid, 20% water, and 79.9% acetonitrile) followed by a column wash. A standard data-dependent acquisition was performed with a full MS spectrum and obtained by the Orbitrap for m/z 400–2000 at the resolution of 120000 with EASY-IC calibration. The precursor ions, with charges from 2–8, were selected, isolated (1.6 m/z window), fragmented by CID, then scanned by the Ion Trap. Survey scans were performed every 2 s, and the dynamic exclusion was enabled for 30 s.

Acquisitions were searched against the NCBInr proteome using PEAKS v10 (Bioinformatics Solutions Inc, Ontario, CA) and a semitryptic search strategy with tolerances of 6 ppm for MS and 0.5 Da for MS/MS, and carbamidomethylation of cysteine as a fixed modification

and oxidation of methionine as a dynamic modification allowing for two missed cleavages. Peptides were filtered with a 0.5% false discovery rate (FDR) using a decoy database approach and a 2 spectral matches/peptide requirement. The LC/MS results for SGE (three sample repeats) and saliva (four sample repeats) were indicated in Supplementary Data 1 and 2, respectively.

## Electron microscopy (EM) of both stained and frozen-hydrated samples

For negative stain EM, 2.5 μL of SGE samples were applied to a glow-discharged grid coated with carbon film. The sample was incubated on the carbon film for 30 s, followed by negative staining with 2% uranyl acetate. Micrographs were recorded on a TIETZ F415MP 16-megapixel CCD camera at a nominal magnification of 70,000× in an FEI Tecnai F20 electron microscope operated at 200 kV.

For cryo-EM sample optimization, an aliquot of 3 μL of sample was applied onto a glow-discharged lacey grid coated with thin continuous carbon (400 mesh, Ted Pella) for 60 s. The grid was blotted with Grade 595 filter paper (Ted Pella) and flash-frozen in liquid ethane with an FEI Mark IV Vitrobot. The same FEI TF20 instrument and imaging condition as negative staining evaluation were used to screen cryo-EM grids. The grids with optimal particle distribution and ice thickness were obtained by varying the gas source (air using PELCO easiGlowTM, target vacuum of 0.37 mbar, target current of 15 mA; or H2/O2 using Gatan Model 950 advanced plasma system, target vacuum of 70 mTorr, target power of 50 W) and time for glow discharge, the volume of applied samples, chamber temperature and humidity, blotting time and force, as well as drain time after blotting. Our best grids were obtained with 30 s glow discharge using air and with the Vitrobot sample chamber set at 8 °C temperature, 100% humidity, 10 s blotting time, 10 blotting force, and 0 s drain time.

Optimized cryo-EM grids were loaded into an FEI Titan Krios electron microscope with a Gatan Imaging Filter (GIF) Quantum LS device and a post-GIF K2 Summit direct electron detector. The microscope was operated at 300 kV with the GIF energy-filtering slit width set at 20 eV. Movies were acquired with SerialEM[39] by electron counting in super-resolution mode at a pixel size of 0.68 Å/pixel (nominal magnification of 105,000×). A total number of 40 frames were acquired in 8 s for each movie, giving a total dose of ~30 e$^-$/Å$^2$/movie.

## Structure determination

Frames in each movie were aligned for drift correction with the GPU-accelerated program MotionCor2[40]. The first frame was discarded during drift correction due to concern of more severe drift/charging of this frame. Two averaged micrographs, one with dose weighting and the other without, were generated for each movie after drift correction. The averaged micrographs have a calibrated pixel size of 1.36 Å at the specimen scale. The averaged micrographs without dose weighting were used only for defocus determination and the averaged micrographs with dose weighting were used for all other steps of image processing.

The defocus value of each averaged micrograph was determined by CTFFIND4[41] to be ranging from -1.5 to -3 μm. Initially, a total of 2,161,624 particles were automatically picked from 2408 averaged micrographs without reference using Gautomatch (https://www2.mrc-lmb.cam.ac.uk/research/locally-developed-software/zhang-software/). The particles were boxed out in dimensions of 300 × 300 square pixels and binned to 150 × 150 square pixels (pixel size of 2.72 Å) before further processing by the GPU accelerated RELION3.0[42]. Several iterations of reference-free 2D classification were subsequently performed to remove "bad" particles (i.e., particles in 2D classes with fuzzy or un-interpretable features, including junk, dissociated particles and contaminations), yielding 714,745 good particles. These particles were subjected to *ab* initio reconstruction with three classes in cryoSPARC v2[43]. One class exhibiting good model features (intact features as

shown in representative 2D classes plus visible secondary structural elements like α-helices) was kept, which was then used as initial model for 3D classification with five classes in RELION3.0. Particles from the best class were re-centered, followed by duplicate removal based on the unique index of each particle given by RELION. The resulting 172,954 particles were un-binned to 300 × 300 square pixels (pixel size of 1.36 Å) and subjected to another round of 2D and 3D classification, yielding 91,280 good particles. Auto-refinement of these particles by RELION generated a map with an average resolution of 3.7 Å.

To gather more particles, another independent data processing pipeline employing similar procedure generated 132,044 good particles. We combined the good particles (91,280 + 132,044), removed duplications (161,375), and performed a final round of 2D classification to clean the dataset. The resulting 161,092 un-binned, unique particles were subjected to a 3D auto-refinement, yielding a map with an average resolution of 3.5 Å. Next, we utilized CTF-refinement in RELION3.0 to estimate beam tilt, asymmetrical aberrations; anisotropic magnification; per-particle defocus values and per-micrograph astigmatism for the entire data set. Subsequently, a final round of 3D auto-refinement was performed in RELION. The two half maps from this auto-refinement step were subjected to RELION's standard post-processing procedure. The final map has an average resolution of 3.3 Å based on RELION's gold-standard FSC. The whole data processing pipeline was summarized in Supplementary Fig. 2a.

## Resolution assessment

All resolutions reported above are based on the "gold-standard" FSC 0.143 criterion[44]. FSC curves were calculated using soft spherical masks and high-resolution noise substitution was used to correct for convolution effects of the masks on the FSC curves[44]. Prior to visualization, all maps were sharpened by applying a negative B-factor which was estimated using automated procedures[45]. Local resolution was estimated using ResMap[46]. The overall quality of the map is presented in Supplementary Fig. 2b-d. Data collection and reconstruction statistics are presented in Supplementary Table 1.

## Sequence identification in the cryo-EM map by *CryoID*

We manually checked the 3.3 Å resolution density map, selected the region with best resolution and build model de novo. The peptide model was then extended on both ends as the density permitted, yielding the following potential sequences, which were then used for searching:

Query Set (corresponding to 2174-2200 and 2612-2636 in the final identified AAEL009993-PA sequence):

1. FSQTYEYVAPGYLADIANNFILEKLLF
2. TGFVMGPDGVLGFYASVGYRVINSA

Using this set of query sequences, *cryoID* identified two candidates for this map from a candidate pool consisting of the top 100 proteins identified in the salivary gland extract by mass spectrometry. These two candidates, AAEL009993-PA and AAEL009992-PA from *Aedes aegypti*, share 53% protein sequence identity. We confirmed the identification by manually building a de novo atomic model into the rest of the map. The map resolution was sufficient for us to confidently determine that the sequence matches AAEL009993-PA rather than AAEL009992-PA based on many of the sidechains which are different between the two proteins, including S2172 and I2263 and Y2659 in AAEL009993-PA.

## Model building and refinement

Following the initial sequence identification using *cryoID*, the SGS1 model in the central regions of the cryo-EM map (ranging from 3.0 to 4.0 Å; Supplementary Fig. 2d) was manually traced and built de novo using COOT[47]. Sequence assignment was mainly guided by visible densities of amino acid residues with bulky side chains, such as Trp,

Tyr, Phe, and Arg. Other residues including Gly and Pro also helped the assignment process. Unique patterns of sequence segments containing such residues were utilized for validation of residue assignment. Secondary structure prediction using PSIPRED 4.0 also guided subsequent model building.

Resolutions for the periphery regions of the cryo-EM map were insufficient for de novo atomic modeling. The following regions were built with neural network-based modeling using Colab AlphaFold2 and rigidly docked into the low-pass filtered map of the 3.3 Å map using CHIMERA[48]: solvent accessible surface of β-propeller 1, residues 913-924 and 999-1045 of the protruding lid of β-propeller 2, residues 2341 to 2447 of the wedge domain. In addition, we tentatively traced the main chain of the stretched loop following β-propeller 1 with residues 345–357.

The above model enabled us to identify extra densities for the predicted TM helices located in the C terminus of SGS1 (residues 2734 to 3042), of which the atomic models were built de novo using COOT with similar methods as mentioned above. No densities for the residues of Tox-SGS were observed in our cryo-EM map, which was consistent with our mass spectrometry result (Supplementary Fig. 4a) and previous report[17] showing no peptides after residue 3035 were recovered from salivary glands in *Aedes aegypti*.

The SGS1 model was refined using PHENIX in real space[49] with secondary structure and geometry restraints. Refinement statistics were summarized in Supplementary Table 1. The model was also evaluated based on Morprobity scores[50] and Ramachandran plots (Supplementary Table 1). Representative densities are shown in Supplementary Fig. 3. All structure-related images in this paper were generated using UCSF CHIMERAX[51] and CHIMERA.

### Phylogenetic tree analysis and multiple sequence alignment

For homologs identification, proteins showing clear sequence similarity to SGS1 of *Aedes aegypti* (Uniprot ID: Q16U82; name: AAEL009993-PA) were collected from the NCBI non-redundant protein database (NCBI nr) using blastp (query Q16U82:1-3364). Hits were retained if they have sequence coverage above 30%. The obtained hits were further verified to remove duplicates, yielding 124 hits. The hit accessions are provided as a Source Data file.

For phylogenetic reconstruction, sequences of the 124 selected hits from various of species were extracted and aligned using MUSCLE (https://www.ebi.ac.uk/Tools/msa/muscle/)[52]. Subsequently, a maximum-likelihood tree was inferred with IQ-TREE (http://iqtree.cibiv.univie.ac.at/)[53] using auto-selected substitution model and 1000 ultrafast bootstrap replicates. The Interactive Tree of Life (iTOL) v6 online tool (https://itol.embl.de/)[54] was used to visualize and annotate the phylogenetic tree, and create the final figure shown as Fig. 1d.

Multiple sequence alignments in Figs. 2c and 3e, and Supplementary Fig. 5 were obtained using MUSCLE and rendered with ESPript 3.0[55].

### Reporting summary

Further information on research design is available in the Nature Portfolio Reporting Summary linked to this article.

## Data availability

The data that support this study are available from the corresponding authors upon reasonable request. Cryo-EM density maps have been deposited in the Electron Microscopy Data Bank (EMDB) under accession numbers EMD-29245 (mosquito salivary gland surface protein 1). Model coordinates have been deposited in the Protein Data Bank (PDB) under accession numbers 8FJP (mosquito salivary gland surface protein 1). Other structures used in this study were obtained from the PDB with accession codes 6H6G (TcB-TcC of Tc-toxin), 6FB3 (Teneurin 2), 6SKA (Teneurin 2 in complex with Latrophilin 1 Lec-Olf domains), 4NOX (nine-bladed beta-propeller of eIF3b), 2XOM

(TmCBM61 in complex with beta-1,4- galactotriose), 3A4U (MCFD2 in complex with carbohydrate recognition domain of ERGIC-53). Protein sequences used in this study were retrieved from Uniprot with accession ID Q16U82 (name AAEL009993-PA) and Q16U81 (name AAEL009992-PA). All other data needed to evaluate the conclusions of this study are present in the paper and/or the supplementary materials. Source data are provided with this paper.

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

## Acknowledgements

We thank Mr. Brian Bonilla for salivary gland dissections and to Drs. Glenn Nardone and Lisa Renee Olano of the Research Technology Branch, NIAID, for mass spectrometry analysis. We also thank Titania Nguyen for editorial assistance. Dr. Adeline E. Williams (NAIID/NIH) thoroughly edited and revised the final version of the manuscript. We acknowledge the use of resources at the Electron Imaging Center for Nanomachines supported by UCLA and grants from the NIH and the National Science Foundation. Our research was supported in part by grants from the U.S. National Institutes of Health (R01GM071940 to Z.H.Z. and Division of Intramural Research Program AI001246 to E.C.). We acknowledge the use of resources at the Electron Imaging Center for Nanomachines of UCLA supported by U.S. NIH (S10RR23057 and S10OD018111) and U.S. NSF (DMR-1548924 and DBI-133813).

## Author contributions

Z.H.Z. and E.C. conceived the project. S.L. and X.X. collected and processed the cryo-EM data and built the atomic models. S.L. prepared the illustrations. S.L, X.X., E.C. and Z.H.Z. interpreted the results. S.L, X.X. and Z.H.Z. wrote the paper. All authors edited and approved the manuscript.

## Competing interests

The authors declare no competing interests.
