## [Peer Review File · Nature Communications]

Native structure of mosquito salivary protein uncovers domains relevant to pathogen transmissionReviewers' Comments:

Reviewer #1:

Remarks to the Author:

The manuscript "Native structure of mosquito salivary protein uncovers domains relevant to pathogen transmission" by Liu et al reports their structural characterization of a large protein called salivary gland surface protein 1 (SGS1) employing cryoID. SGS1 is one of the 100+ proteins present in mosquito saliva and one of the most abundant ones in the pool. With more than 3,000 residues and poorly defined structures based on its primary sequence, the real function of SGS1 and how its function is exerted in the cell have remained largely unknown. The work reported in this manuscript represents a major step forward to shedding light on the structure and potential function of SGS1, particularly because the structure were determined in the native state of the protein without any perturbation of its expression and in vivo localization.

The authors have determined the cryo-EM structure at an overall resolution of 3.3 Å for a nearly full-length SGS1 isolated directly from the salivary gland of mosquitos. The structure reveals a cocoon-like core structure formed by the complex Rhs/YD repeats, which bears a spacious chamber inside and is decorated by five other globular domains on its surface. Despite there are some similarities in the central core structure they determined for SGS1 in comparison to several other well characterized proteins such as Tc toxin and Teneurin, their work still provides much more information about SGS1 beyond that. One of the biggest surprises is that the C-terminal region of SGS1 with ~230 residues, which was predicted to form a multi-helical TM domain, turned out to be nearly fully buried in the chamber of the cocoon shell. Their MS data confirmed this buried C-terminal region was cleaved off from the preceding sequence by proteolysis when SGS1 is secreted to saliva.

Overall, the work presented in this manuscript shows several novel findings about SGS1 and would be very important to the fields of both structural biology and parasitology. The article was written properly, and the reported findings are interesting in many aspects. The cryoID method used in their studies is a technique originally developed by their group and has proved very powerful in characterizing large biological assemblies in their native state. I only have a few minor questions as listed below.

1. They predicted that the putative TM segments buried in the cocoon chamber will be released upon the injection of mosquito saliva into human bodies. Could they think of any possible regulation for that? For example, would pH change or other cues in the body trigger the release of such extensively and strongly bound long polypeptides (most hydrophobic) from the chamber? Do they think there could potentially be some dramatic rearrangement of the cocoon structure to "squeeze" out the TM regions?

2. Their data showed that the Tox-SGS domain at the very C-terminus of SGS1 is already cleaved off at a very early stage as this part was absent even in the sample they examined from the salivary gland. Are there any literatures reporting what the function of the Tox-SGS is? If yes, they should include these early reports in their manuscript. If not, could they suggest any potential function for the Tox-SGS domain of SGS1?

3. Both PDB code and EMD code are missing and should be included if their data depositions have already been done.

Reviewer #2:

Remarks to the Author:

In this manuscript, Liu et al present a high-quality structural analysis of the SGS1 protein from the salivary gland of a mosquito. This was determined without protein purification as the major large protein in the salivary glands, with its identification confirmed using BioID.

The structure determination appears to be of high-quality, although without seeing a validation report, which I didn't see in the files, it is not possible to be totally sure. These validation reports should be provided.

The structure provides novel insight into the architecture of this protein, showing a core of Rhs/YD shells, decorated with four additional domains, which are each of folds commonly associated with protein-protein or protein-glycan interactions. Inside the shell are found the putative transmembrane helices, revealing an interesting possible mechanism why which such a protein could have both soluble and membrane associated forms.

In general, this is a high quality piece of work and describes the novel structure of an important protein. There is no functional data presented here, to probe the role of the protein, or the functions of its different domains. However, the structure will be very valuable to the field, opening opportunities to conduct such experiments in the future. The work is well done and of substantial value.

Itemized Responses to Reviewers' Comments: NCOMMS-22-40382-T

Summary of responses: We thank both reviewers for the careful and supportive reviews, and the editor for the interest in publishing our paper. As you will see from the itemized responses below, we have addressed the issues raised by the reviewers and revised the manuscript accordingly. To facilitate your navigation of this document, we have copied the reviewers' comments verbatim in **black** and our responses are shown in **blue**.

Reviewer #1 (Remarks to the Author):

The manuscript "Native structure of mosquito salivary protein uncovers domains relevant to pathogen transmission" by Liu et al reports their structural characterization of a large protein called salivary gland surface protein 1 (SGS1) employing cryoID. SGS1 is one of the 100+ proteins present in mosquito saliva and one of the most abundant ones in the pool. With more than 3,000 residues and poorly defined structures based on its primary sequence, the real function of SGS1 and how its function is exerted in the cell have remained largely unknown. The work reported in this manuscript represents a major step forward to shedding light on the structure and potential function of SGS1, particularly because the structure was determined in the native state of the protein without any perturbation of its expression and in vivo localization.

The authors have determined the cryo-EM structure at an overall resolution of 3.3 Å for a nearly full-length SGS1 isolated directly from the salivary gland of mosquitos. The structure reveals a cocoon-like core structure formed by the complex Rhs/YD repeats, which bears a spacious chamber inside and is decorated by five other globular domains on its surface. Despite there are some similarities in the central core structure they determined for SGS1 in comparison to several other well characterized proteins such as Tc toxin and Teneurin, their work still provides much more information about SGS1 beyond that. One of the biggest surprises is that the C-terminal region of SGS1 with ~230 residues, which was predicted to form a multi-helical TM domain, turned out to be nearly fully buried in the chamber of the cocoon shell. Their MS data confirmed this buried C-terminal region was cleaved off from the preceding sequence by proteolysis when SGS1 is secreted to saliva.

Overall, the work presented in this manuscript shows several novel findings about SGS1 and would be very important to the fields of both structural biology and parasitology. The article was written properly, and the reported findings are interesting in many aspects. The cryoID method used in their studies is a technique originally developed by their group and has proved very powerful in characterizing large biological assemblies in their native state. I only have a few minor questions as listed below.

Response: We are grateful to this reviewer for her/his very nice summary on the major novelty of our work and its significance to the field.

1. They predicted that the putative TM segments buried in the cocoon chamber will be released upon the injection of mosquito saliva into human bodies. Could they think of any possible regulation for that? For example, would pH change or other cues in the body trigger the release of such extensively and strongly bound long polypeptides (most hydrophobic) from the chamber? Do they think there could potentially be some dramatic rearrangement of the cocoon structure to "squeeze" out the TM regions?

Response: We currently don't have evidence on what triggers the release of putative TM segments upon the injection of mosquito saliva. Possibilities include pH changes, host cofactors-binding and/or membrane attachment. We agree that rearrangement near the middle opening of cocoon structure (such as movement of

the "fence" shown in panel iv of Fig. 2a) may be needed to generate enough space to free partially folded TM helices and is so indicated in the revised manuscript (see lines 262-265).

2. Their data showed that the Tox-SGS domain at the very C-terminus of SGS1 is already cleaved off at a very early stage as this part was absent even in the sample they examined from the salivary gland. Are there any literatures reporting what the function of the Tox-SGS is? If yes, they should include these early reports in their manuscript. If not, could they suggest any potential function for the Tox-SGS domain of SGS1?

Response: So far, there is no literature reporting the function of Tox-SGS domain. Our protein sequence analysis indicated Tox-SGS is a conserved, arthropod-specific domain, but didn't reveal any functional relevance; our recombinant overexpression attempts of this domain has not been successful; structure prediction via Alphafold didn't generate meaningful 3D models with high confident score for this domain. The early cleavage of Tox-SGS domain prior to being secreted into salivary gland suggests Tox-SGS may function in mosquito hemolymph or other tissues encountered by SGS1.

3. Both PDB code and EMD code are missing and should be included if their data depositions have already been done.

Response: We have not obtained the accession codes of SGS1 from PDB (8FJP) and EMD (EMD-29245). These codes have been added in the revised manuscript.

Reviewer #2 (Remarks to the Author):

In this manuscript, Liu et al present a high-quality structural analysis of the SGS1 protein from the salivary gland of a mosquito. This was determined without protein purification as the major large protein in the salivary glands, with its identification confirmed using BioID. The structure determination appears to be of high-quality, although without seeing a validation report, which I didn't see in the files, it is not possible to be totally sure. These validation reports should be provided.

Response: As indicated above in our response to Reviewer #1, we have deposited our cryo-EM map and atomic model to EMD and PDB database and have now obtained from them official accession codes, EMD-29245 and 8FJP, respectively. The validation report is included as part of the revised manuscript submission.

The structure provides novel insight into the architecture of this protein, showing a core of Rhs/YD shells, decorated with four additional domains, which are each of folds commonly associated with protein-protein or protein-glycan interactions. Inside the shell are found the putative transmembrane helices, revealing an interesting possible mechanism why which such a protein could have both soluble and membrane associated forms.

In general, this is a high-quality piece of work and describes the novel structure of an important protein. There is no functional data presented here, to probe the role of the protein, or the functions of its different domains. However, the structure will be very valuable to the field, opening opportunities to conduct such experiments in the future. The work is well done and of substantial value.

Response: We thank you for recognizing the novelty and significance of this paper. We agree that future work will be necessary to understand how SGS1 facilitates *Plasmodium* sporozoite/arbovirus invasion of the salivary gland and/or modulating the host's immune response. The availability of the first atomic model of this extremely large protein enabled by this study now opens the door for engineer mutant strains or recombinant constructs for functional studies, which should help clarify the specific roles of SGS1 domains in these processes.

Reviewers' Comments:

Reviewer #1:

None

Reviewer #2:

Remarks to the Author:

I had no suggestions for changes to the first version of the manuscript other than that the validation reports should have been provided. They now have and I have checked them. The quality of the structure appears high and I am happy to recommend publication.

Congratulations to the authors on a nice study.